# Predicting SARS-CoV-2 Variant Spread in a Completely Seropositive Population Using Semi-Quantitative Antibody Measurements in Blood Donors

**DOI:** 10.3390/vaccines10091437

**Published:** 2022-08-31

**Authors:** Lewis Buss, Carlos A. Prete, Charles Whittaker, Tassila Salomon, Marcio K. Oikawa, Rafael H. M. Pereira, Isabel C. G. Moura, Lucas Delerino, Rafael F. O. Franca, Fabio Miyajima, Alfredo Mendrone Jr., Cesar Almeida-Neto, Nanci A. Salles, Suzete C. Ferreira, Karine A. Fladzinski, Luana M. de Souza, Luciane K. Schier, Patricia M. Inoue, Lilyane A. Xabregas, Myuki A. E. Crispim, Nelson Fraiji, Luciana M. B. Carlos, Veridiana Pessoa, Maisa A. Ribeiro, Rosenvaldo E. de Souza, Anna F. Cavalcante, Maria I. B. Valença, Maria V. da Silva, Esther Lopes, Luiz A. Filho, Sheila O. G. Mateos, Gabrielle T. Nunes, David Schlesinger, Sônia Mara Nunes da Silva, Alexander L. Silva-Junior, Marcia C. Castro, Vítor H. Nascimento, Christopher Dye, Michael P. Busch, Nuno R. Faria, Ester C. Sabino

**Affiliations:** 1MRC Centre for Global Infectious Disease Analysis, Imperial College London, London SW7 2BX, UK; 2Department of Electronic Systems Engineering, University of São Paulo, São Paulo 05508-010, Brazil; 3Faculdade Ciências Médicas de Minas Gerais, Belo Horizonte 30130-110, Brazil; 4Center of Mathematics, Computing and Cognition, Universidade Federal do ABC, Santo André 09210-170, Brazil; 5Institute for Applied Economic Research (Ipea), Brasília 70390-025, Brazil; 6Fundação Oswaldo Cruz, Rio de Janeiro 21040-900, Brazil; 7Universidade Federal do Ceará, Fortaleza 60455-760, Brazil; 8Fundação Pró Sangue Hemocentro de São Paulo (FPS), Sao Paulo 05403-000, Brazil; 9Centro de Hematologia e Hemoterapia do Paraná (HEMEPAR), Curitiba 80045-145, Brazil; 10Fundação Hospitalar de Hematologia e Hemoterapia do Amazonas (HEMOAM), Manaus 69050-001, Brazil; 11Centro de Hematologia e Hemoterapia do Ceará (HEMOCE), Fortaleza 60140-200, Brazil; 12Fundação Hemominas, Belo Horizonte 30150-341, Brazil; 13Fundação de Hematologia e Hemoterapia de Pernambuco (HEMOPE), Recife 52011-000, Brazil; 14Instituto Estadual de Hematologia Arthur de Siqueira Cavalcanti (HEMORIO), Rio de Janeiro 20211-030, Brazil; 15Mendelics, São Paulo 02511-000, Brazil; 16Universidade Federal do Amazonas, Manaus 69067-005, Brazil; 17Harvard T.H. Chan School of Public Health, Boston, MA 02115, USA; 18Department of Zoology, University of Oxford, Oxford OX1 3SZ, UK; 19Vitalant Research Institute, Denver, CO 80230, USA; 20Department of Laboratory Medicine, University of California San Francisco, San Francisco, CA 94158, USA; 21Departamento de Moléstias Infecciosas e Parasitárias e Instituto de Medicina Tropical da Faculdade de Medicina, Universidade de São Paulo, São Paulo 05508-010, Brazil; 22Department of Infectious Disease Epidemiology, Imperial College London, London SW7 2BX, UK

**Keywords:** SARS-CoV-2, seroprevalence, variants of concern, immunity, vaccines, delta

## Abstract

SARS-CoV-2 serologic surveys estimate the proportion of the population with antibodies against historical variants, which nears 100% in many settings. New approaches are required to fully exploit serosurvey data. Using a SARS-CoV-2 anti-Spike (S) protein chemiluminescent microparticle assay, we attained a semi-quantitative measurement of population IgG titers in serial cross-sectional monthly samples of blood donations across seven Brazilian state capitals (March 2021–November 2021). Using an ecological analysis, we assessed the contributions of prior attack rate and vaccination to antibody titer. We compared anti-S titer across the seven cities during the growth phase of the Delta variant and used this to predict the resulting age-standardized incidence of severe COVID-19 cases. We tested ~780 samples per month, per location. Seroprevalence rose to >95% across all seven capitals by November 2021. Driven by vaccination, mean antibody titer increased 16-fold over the study, with the greatest increases occurring in cities with the highest prior attack rates. Mean anti-S IgG was strongly correlated (adjusted R2 = 0.89) with the number of severe cases caused by Delta. Semi-quantitative anti-S antibody titers are informative about prior exposure and vaccination coverage and may also indicate the potential impact of future SARS-CoV-2 variants.

## 1. Introduction

Serologic surveys estimate the proportion of a population with detectable antibodies against SARS-CoV-2, to infer cumulative incidence of infection or vaccination coverage (e.g., [1]). This approach was useful early in the COVID-19 pandemic to estimate approximate total infections, important for epidemic modelling and determining fatality ratios. However, as population exposure to SARS-CoV-2 antigens through infection or vaccination reaches 100%, so too will seroprevalence (e.g., [2]), making this indicator meaningless. Interpretation is further limited by the successive emergence of SARS-CoV-2 variants of concern (VOC) [3] with increasing transmissibility [4,5] and partial immune escape [6,7], which continue to cause epidemics in populations with high documented immunologic exposure [8,9].

The semi-quantitative output of serological assays, reported as the ratio of signal to cut-off (S/C), arbitrary units (AU)/mL, or binding antibody units (BAU)/mL, reflects IgG or total antibody titers and thus provides more information than a binary positive versus negative interpretation. The average population S/C value will be some function of the extent and timing of prior natural infection, as well as vaccine coverage, type and date. Together, these factors will shape population patterns of immunity towards SARS-CoV-2 infection and severe COVID-19 disease [10]. Previous work has shown neutralizing antibody levels to be highly predictive of protection from SARS-CoV-2 symptomatic infection [11]. As such, higher average semi-quantitative anti-S levels might predict a lower incidence of severe cases caused by a new variant introduced into a seropositive population.

Here, we test this hypothesis using serial, cross-sectional, semi-quantitative, anti-Spike (S) protein measurements from across Brazil in 2021 when the Gamma [6], Delta [12] and Omicron [13] VOCs successively replaced one another. We assess the extent to which vaccination and prior infection contributed to measured population anti-S IgG levels, and the degree to which these variables predicted the incidence of severe COVID-19 cases caused by the Delta VOC.

## 2. Materials and Methods

### 2.1. Study Design and Blood Donor Sampling Strategy

The blood donor sampling strategy has previously been reported in detail [9,14]. Briefly, we aimed to test 850 blood donation samples each month from public blood banks in the seven participating Brazilian cities (São Paulo, Rio de Janeiro, Manaus, Recife, Fortaleza, Curitiba, Belo Horizonte). Starting from the second week of each month, we selected consecutive samples among all donations (in Manaus) or within city neighborhoods to achieve spatial representativity (remaining cities). Sample selection proceeded until quotas were filled or the available samples were exhausted. Sampling spanned donations from April 2021 to November 2021 in all seven cities. In addition, we tested samples from November 2020 in Manaus (previously reported in [14]) and March 2021 in Recife.

### 2.2. SARS-CoV-2 Serology Assay

We used a chemiluminescent microparticle immunoassay (CMIA, AdviseDx, Abbott, Chicago, USA) that detects IgG antibodies against the SARS-CoV-2 spike (S) protein. At a threshold of 50 S/C units, we have previously shown this assay to have a sensitivity of 94.0% in 208 non-hospitalized PCR-positive convalescent individuals, and its specificity is >99% [15].

### 2.3. Secondary Data Sources

Data for vaccine doses administered were taken from the OpenDataSUS [16]. Population denominators were projected estimates for 2021 based on the 2010 Brazilian census [17].

We analyzed three different sources for COVID-19 cases. First, we retrieved all severe acute respiratory syndrome cases (SARI, confirmed COVID-19 and unknown etiology) and deaths from the SIVEP-Gripe [16]. Total SARI is less affected by underreporting [18,19] but only reflects severe cases. Second, we retrieved the total number of confirmed COVID-19 cases (irrespective of severity) reported by the Brazilian Ministry of Health. The official MoH case counts should be treated with caution as they are heavily influenced by access to testing and are published by date of reporting, not date of symptom onset [18]. Finally, we further obtained the time series of SARS-CoV-2 test positivity in a network of pharmacies in São Paulo city (https://mendelics.com.br/).

To determine the relative abundance of SARS-CoV-2 variants, we retrieved metadata from all SARS-CoV-2 genomes deposited on GISAID [20] between March 2020 and March 2022 in the seven states. We grouped SARS-CoV-2 lineages into Omicron, Delta, Gamma and wildtype, and summarized weekly lineage counts per location.

### 2.4. Data Analysis and Statistics

We fitted a multinomial model to weekly variant counts with calendar time as the predictor variable and a two-knot cubic spline [4] using the nnet package [21] in R (version 4.1.2, R Foundation for Statistical Computing, Vienna, Austria). Using this model, we arbitrarily defined periods of variant dominance in each state as beginning from the week when each variant first reached 10% relative abundance.

We calculated the incidence of SARI cases and deaths using the projected population size for the seven cities. In the total case data from the MoH, there was an artefactual increase in the number of reported cases in a small number of weeks, reflecting changes in reporting, not transmission. To remove this effect but preserve the overall shape of the epidemic curve, we excluded four weeks for Curitiba and one for Recife where the case counts were > 10 times the median for those cities.

We first calculated monthly IgG anti-S seroprevalence estimates with exact binomial 95% confidence intervals (95% CI) for all cities, using the manufacturers’ threshold of 50 S/C units to define a positive reading. We then calculated the geometric mean of the semi-quantitative anti-S IgG S/C readings from all blood donations for each month and location. S/C scales vary between serology assays, and there is no direct biological interpretation of these values. As such, we followed the approach of Khoury et al. and Earle et al., [11,22] and standardized the mean anti-S IgG S/C values against a convalescent cohort. We calculated the ratio of the mean monthly S/C readings to the mean S/C value seen in a cohort of 245 convalescent samples collected following wildtype infection in 2020. This cohort has been described in detail previously [9,14,23].

In primary studies of vaccine efficacy, the convalescent-normalized mean anti-S S/C induced post-vaccination correlate strongly with protection against PCR-confirmed symptomatic infection with wildtype SARS-CoV-2 [11,22]. We fitted a linear regression to results from seven primary vaccine studies (data from [22]), and used this to estimate convalescent-normalized mean anti-S S/C that equate to 65%, 75% and 85% vaccine efficacy.

We next explored the association between convalescent-normalized anti-S S/C, vaccination and prior attack rate. We exploited the variation in vaccination timing across the blood donor age range (15–69 years), in which older individuals were vaccinated before younger individuals. We built a multivariate linear regression of monthly convalescent-normalized mean S/C in 10-year age groups on vaccine coverage with the first dose, second dose and prior attack rate by December 2020 (as estimated previously [14]) in each city. We built a null model containing all three variables and no interaction terms. We then added interaction terms between attack rate and first dose coverage, and attack rate and second dose coverage. We selected the best performing model using the Bayesian Information Criterion.

Finally, we assessed the predictive value of convalescent-normalized mean anti-S S/C to predict the spread of the Delta VOC across the seven cities. For a two-month period starting from the day Delta reached 10% dominance in each city, we calculated age-standardized SARI incidence in the blood donor age range. Standardization was with the direct method, taking the population of São Paulo as the reference. We then regressed age-standardized SARI incidence on vaccine coverage (first and second dose separately) and convalescent-normalized mean anti-S S/C.

## 3. Results

### 3.1. SARS-CoV-2 Infection, Vaccination and Seroconversion across Seven Brazilian Capitals

The seven capital cities (Belo Horizonte, Curitiba, Fortaleza, Manaus, Recife, Rio de Janeiro and São Paulo) are located across four of Brazil’s five macro-regions (Figure 1A). Vaccination was rolled out across the seven cities at the beginning of 2021. Among residents in the blood-donation-eligible age range (15–69 years), coverage with the first dose reached > 75% across all cities by the end of the study period. Coverage with the second dose was also high by this point (Figure 1B). The share of vaccine types is shown in Figure 1C. Importantly, Sinovac, which induces both anti-S and anti-N antibodies. accounted for ~25% across all cities. For this reason, the presence of anti-N antibody cannot be used to distinguish natural infection from vaccine-induced seroconversion in the Brazilian context.

Distinct SARS-CoV-2 epidemics, in both shape (Figure 1D) and size (Figure 1E), have occurred across these cities. For example, the cumulative attack rate (inferred from seroprevalence) in December 2020, prior to the Gamma-driven second wave and prior to the rollout of vaccination campaigns in Brazil, ranged from 20.3% (95% confidence interval, 95% CI, 18.6% to 22.3%) in Curitiba, to 76.3% (95% CI 72.1% to 81.4%) in Manaus (Figure 1E) [14].

While all cities experienced a large Gamma-dominated peak in cases and deaths (Figure 1D, Appendix A), the subsequent period of Delta dominance was not marked by similarly significant epidemics. Indeed, following Delta’s introduction, the incidence of cases and deaths was persistently low (Manaus, Fortaleza, Recife), falling (Belo Horizonte, São Paulo) or falling with a small initial increase (Rio de Janeiro, Curitiba); see (Figure 1D). There was a variable peak in the number of reported cases with the introduction of Omicron, but a negligible increase in deaths: only 3.7% of deaths were reported in this period.

We tested on average 780 (range 247–997) blood donation samples (per month, per location) for anti-S IgG. The proportion of donors testing positive on this assay (S/C > 50) increased steadily during 2021, reaching > 95% in all cities (95.1% in Fortaleza to 98.6% in Recife) by November 2021.

### 3.2. Convalescent-Normalized Mean S/C in Blood Donors, Vaccination Coverage and Prior Attack Rate

The monthly convalescent-normalized mean anti-S readings are presented in Figure 2A (raw data in Appendix A). S/C values corresponding to 65%, 75% and 85% hypothetical vaccine efficacy against wildtype are shown as dashed lines (Figure 2). Owing to several limitations (see Discussion), these should not be interpreted as the true population protection in the cities at these time points.

Mean antibody concentration increased 16-fold between May and August 2021 (Figure 2A). During this period, the first vaccine doses were administered in the blood donor age range (Figure 1B). This appears to have been the proximal factor driving the dramatic increase in antibody S/C values. Coincident with this, a variable amount of natural infection occurred across the seven cities (Figure 1D), and the contribution of the two cannot be completely separated.

Prior attack rate influenced the S/C reached when vaccine coverage was at 50% (Figure 2B,C). In Manaus, where the prior attack rate was highest, administration of the first dose was associated with a large increase in antibody levels (Figure 2B), whereas the incremental change following the second dose was small (Figure 2C). In the other cities, the second dose was associated with larger increases in antibody titers. Consistent with these observations, the best-fitting model for convalescent-normalized antibody level included interactions between prior attack rate and first and second dose coverage (Table 1). Assuming a prior attack rate of zero, each 10% increase in first dose vaccination resulted in a 1.17-fold (95% CI, 1.09 to 1.26) increase in convalescent-normalized mean antibody levels. The magnitude of this increase was 1.03 (1.01–1.04) times greater for each 10% increase in the prior attack rate. By contrast, the effect of the second dose coverage was 0.97 (0.95–0.99) smaller for each 10% increase in the prior attack rate.

### 3.3. Anti-S IgG Levels in Blood Donors as Predictor of Delta’s Epidemic Penetrance

The Delta VOC reached a 10% share of genomes on GISAID first in Curitiba (19 June 2021) and last in Manaus (16 August 2021). There was a strong relationship between age-standardized SARI and anti-S S/C (Figure 3), with 89% of the variance in epidemic size explained. A weaker relationship existed between population coverage with the first and second vaccine doses (Figure 3), with 49% and 39% of the variance explained by these variables, respectively.

## 4. Discussion

We present serial cross-sectional anti-S IgG results in blood donors across seven Brazilian state capitals in 2021. Nearly all donors were seropositive by the end of the year. While vaccination was a key factor that increased quantitative antibody levels, historical attack rates determined the dynamics and scale of these increases. Specifically, first-dose coverage was associated with a steeper increase in antibody S/C in cities with higher prior attack rates. By contrast, the second dose was associated with a smaller increment in antibodies when prior attack rate was higher. This is consistent with individual-level data [24] showing that the first vaccination dose induces high anti-S S/Cs in individuals previously infected by SARS-CoV-2, and that these levels were similar to double-dosed immune naïve people. By contrast, the second dose produced a minimal increment in antibody level among convalescents but had a larger effect on those not previously infected. Our results show that semi-quantitative anti-S levels in blood donors varied as a function of the complex combination of immunizing events within these seven Brazilian urban populations.

An important question is whether average population anti-S S/C readings have an epidemiologic interpretation—specifically, whether they can they be used to predict morbidity from emerging variants. This might be expected to be the case, given the strong relationship between mean anti-S binding antibody level induced by vaccination and the group-level protection against wildtype infection and COVID-19 disease penetrance [22]. Furthermore, anti-S levels correlate strongly with neutralizing antibody titers [23], and these in turn are predictive of group-level vaccine efficacy [11,22]. Indeed, our results showed that, during the growth phase of the Delta VOC in Brazil, the incidence of SARI cases was strongly correlated with antibody levels measured in blood donors.

This finding is further supported by individual-level data [25]. The risk of infection with the Delta VOC has been shown to be lowest in people who had experienced prior infection and vaccination [25]. Prior infection alone conferred greater protection than vaccination alone, but both were inferior to hybrid immunity. Given the high level of both prior infection and vaccination coverage in some Brazilian cities, most notably Manaus, it is not surprising that the Delta VOC caused a negligible number of severe cases in these locations (Figure 1D and Figure 3). In Brazil, where vaccination coverage is high and with great spatial heterogeneity in prior attack rates, it follows that a marker that reflects immunity from both infection and vaccination (i.e., mean anti-S S/C) is predictive of Delta’s spread. However, due to limitations of the ecological study design, causality cannot be shown.

By November 2021, mean anti-S titers were similarly high across all seven cities. With the introduction of Omicron, there was a large spike in total cases across all cities. However, there was only a modest peak in SARI cases and negligible numbers of deaths. The exception to this was Fortaleza, where a large increase in SARI cases attributable to Omicron infections was observed. However, the decoupling of SARI cases from deaths suggests this was a reporting artifact and not a true reflection of higher morbidity. Therefore, our results show that antibodies, acquired against historical variants (infection and vaccination), were sufficiently high in Brazil to prevent a significant public health impact from the Omicron VOC. However, due to homogeneity in high mean antibody levels across the cities by the time of the Omicron wave, we could not repeat the analysis performed for Delta (Figure 3). If antibody titer data were to be shared by a variety of global locations, the relationship between antibody levels and public health impact (severe cases and deaths) of new variants could potentially be assessed in near real-time and contribute to more informed policy decisions.

There are several limitations to this work. First, we have previously argued that our blood donor samples are representative of the wider population with respect to SARS-CoV-2 infection [6,9,14]. It is less clear to what extent blood donors reflect population vaccination coverage. We can speculate that individuals who donate blood are less likely to be vaccine hesitant than those who do not donate, as blood donation involves engaging with traditional healthcare services.

Inconsistencies between cities in total reported cases, severe cases (SARI), deaths and attack rate estimation based on blood donor serosurveillance (Figure 1D,E) highlight the ongoing challenges in real-time epidemic monitoring in Brazil. For example, the incidence of total reported cases (Brazilian Ministry of Health) during the Omicron-driven wave varied greatly between cities, with Rio de Janeiro and Fortaleza showing exceptionally large peaks, compared to São Paulo, where rates increased only slightly. Using alternative metrics of epidemic development may be of value; for example, Appendix A shows PCR test positivity in a large network of pharmacies located in São Paulo city. During most of the pandemic, test positivity tracked closely with reported cases; however, in the Omicron-dominated peak, official reported cases appear to have severely underestimated the true magnitude of the epidemic wave. Caution is needed when interpreting case data, particularly for this period in Brazil. Furthermore, vaccination status is not routinely recorded in aggregate case data, precluding a more nuanced analysis including this variable.

Two issues in particular are relevant to the application of our results in other settings. First, our study was conducted during vaccine roll-out in Brazil, and therefore, the findings reflect population antibody titer and immunity before significant waning had occurred. The correlation between protective immunity and antibody titer during the waning phase remains to be assessed at either individual or population levels. If these variables do remain strongly correlated, then monitoring of population antibody titer may be informative about timing of booster doses, for example. Second, it is unclear how mean antibody titers will predict protection against current or future variants that continue to accumulate mutations associated with immune escape and increased fitness [26]. It seems reasonable to assume that mean titers of binding antibodies developed against historical variants would remain correlated with morbidity caused by contemporary or future variants, even if the parameters that describe the relationship are different. For this reason, it is important that serosurveys share antibody titer data, and not simply seroprevalence estimates, so this question can be addressed.

## Figures and Tables

**Figure 1 vaccines-10-01437-f001:**
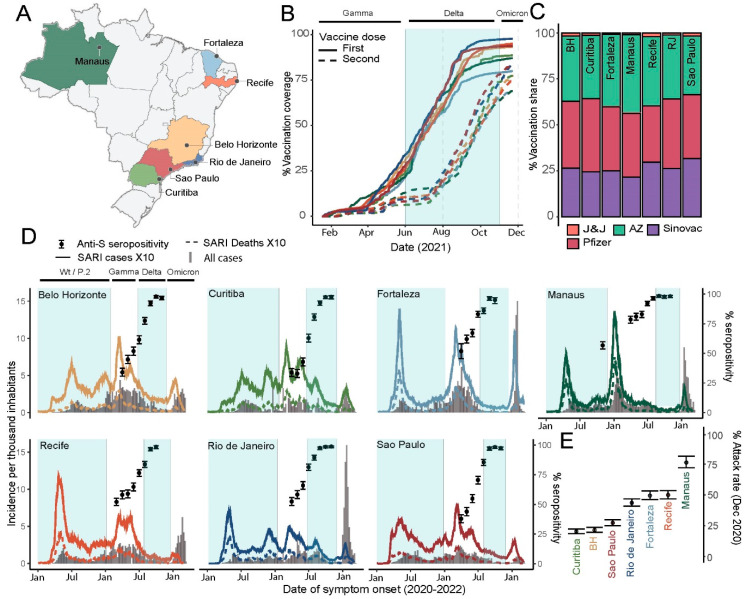
SARS-CoV-2 infection and vaccination across seven Brazilian capital cities. (**A**) Location of the seven state capitals with participating blood donation centers. (**B**) Proportion of the population within the donation-eligible age group (15–65 years) having received one or two doses of SARS-CoV-2 vaccine in each of the seven cities. (**C**) The relative share of vaccine types cumulative through December 2021. BH—Belo Horizonte, RJ—Rio de Janeiro, J&J—Johnson and Johnson, AZ—AstraZeneca. Vaccination data were extracted from OpenDataSUS (https://opendatasus.saude.gov.br/). (**D**) Incidence of cases and deaths due to severe acute respiratory syndrome (SARI, data are from SIVEP-Gripe national reporting system) multiplied by 10 (for visualization alongside total cases) and total cases reported by the Brazilian Ministry of Health (https://covid.saude.gov.br/). Periods of variant dominance start from the date at which each variant reached a 10% share of all sequences deposited on GISAID (https://www.gisaid.org/), based on predictions from a multinomial model fit to these data (Appendix A). Anti-S seropositivity is calculated based on monthly blood donor samples and is shown with exact 95% binomial confidence intervals (error bars). (**E**)—estimated cumulative attack rate in December 2020 based on anti-N serosurveillance in the same blood donor population [14]. 95% confidence intervals are shown as error bars.

**Figure 2 vaccines-10-01437-f002:**
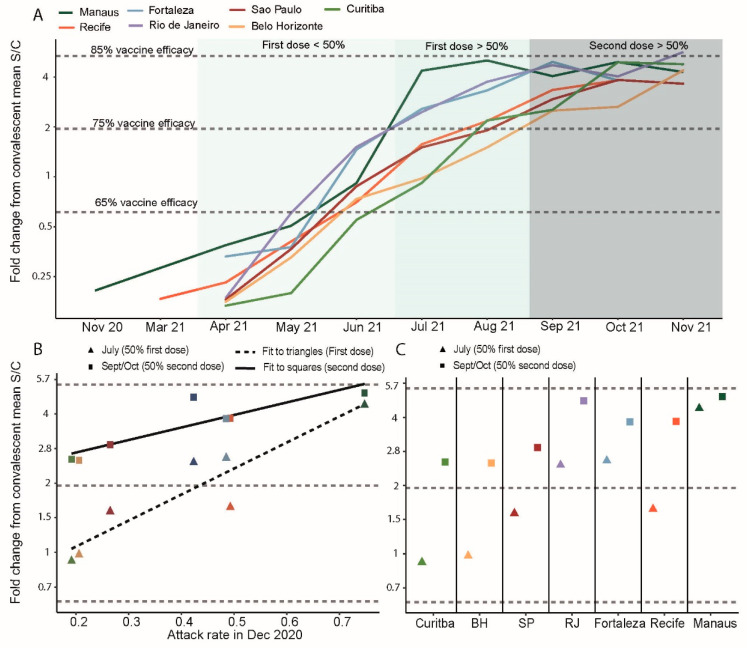
Monthly convalescent-normalized mean antibody titer in blood donors across seven Brazilian cities. (**A**) Mean anti-S IgG antibody titer in blood donors normalized against mean convalescent anti-S IgG level shortly following infection. (**B**) Convalescent-normalized anti-S IgG when first dose coverage reached 50% in all cities (July 2021, triangles) and second dose vaccination reached 50% (September or October, squares) against estimated attack rate in December 2020 [14]. (**C**) Symbols as in B but grouped by city on x-axis. Vaccine efficacy estimates (horizontal dashed lines) are based on the relationship between convalescent-normalized mean anti-S IgG following vaccine administration and protection against PCR-confirmed symptomatic infections (as described in [22]).

**Figure 3 vaccines-10-01437-f003:**
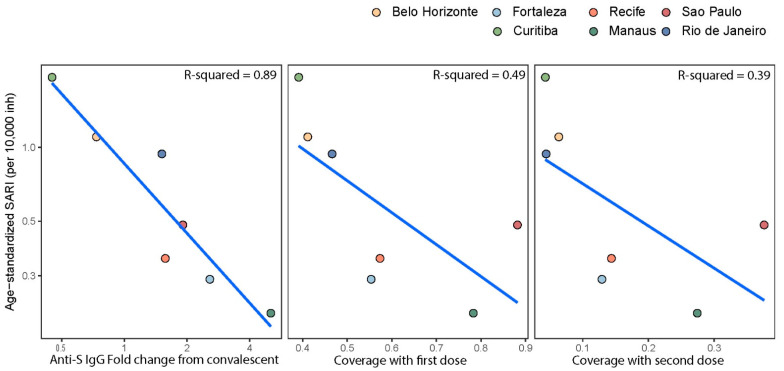
Predictors of epidemic size of the Delta growth phase across seven Brazilian cities. Convalescent-normalized mean anti-S IgG signal to cut-off was calculated for the month when Delta reached 10% dominance in each of the cities (range 19 June 2021 in Curitiba to 16 August 2021 in Manaus). Percentage coverage with the first and second doses was also calculated up to (and inclusive of) the month of 10% dominance in each city. Total severe acute respiratory syndrome (SARI) cases, within the age range of blood donors (15–65 years) were age-standardized using the direct method and the age structure of São Paulo as the reference population. A two-month period starting from the date of 10% dominance was used to calculate epidemic size. R-squared terms are from separate simple linear models fit to the seven points shown on the figure.

**Table 1 vaccines-10-01437-t001:** Best-performing linear regression model of convalescent-normalized monthly antibody anti-S titers.

Model Terms Per Capita × 10 ^1^	Point Estimate (95% CI) Fold Change form Mean Convalescent	*p*-Value
First dose coverage	1.18 (1.10–1.26)	<0.001
Second dose coverage	1.15 (1.07–1.24)	<0.001
Attack rate	1.07 (0.99–1.16)	0.10
First dose × attack rate	1.03 (1.01–1.05)	0.002
Second dose × attack rate	0.97 (0.95–0.99)	<0.001

^1^ An increase of 1 unit corresponds to a 10% increase in coverage or attack rate.

## Data Availability

Continuing our previous serological data sharing initiative [8], all raw serological data are available at https://github.com/CADDE-CENTRE.

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
