# Peer review of "Predicting SARS-CoV-2 Variant Spread in a Completely Seropositive Population Using Semi-Quantitative Antibody Measurements in Blood Donors"

_vaccines, 2022, doi:10.3390/vaccines10091437_

Round 1

Reviewer 1 Report

The results and conclusion are not impressive and the conclusions are as expected without this study:

1.       The booster vaccine dose is more effective in case of absence of COVID-19 infection after 1st injection.

2.       The average antibody level measurement may help to determine the time for booster dose.

3.       The antibody titers reflect the immunity level of the population but does not show the level of protection against future variants of coronavirus.

 So, two of three these conclusions are obvious. My opinion is that the manuscript data are useful but not very important because a lot of similar data were published earlier. And I do not see how the authors may improve this manuscript without adding some important data about morbidity among non-vaccinated and vaccinated persons. It would be useful to publish it because these data are from South American country but not very important.

Author Response

Response to reviewer 1

Query 1.

The results and conclusion are not impressive and the conclusions are as expected without this study:

  1. The booster vaccine dose is more effective in case of absence of COVID-19 infection after 1stinjection.
  2. The average antibody level measurement may help to determine the time for booster dose.
  3. The antibody titers reflect the immunity level of the population but does not show the level of protection against future variants of coronavirus.

 So, two of three these conclusions are obvious. My opinion is that the manuscript data are useful but not very important because a lot of similar data were published earlier.

Response 1

We thank the reviewer for recognizing the incremental value, building on existing scientific findings, that our data bring. We also would like to highlight that we are sharing these data publicly at our data repository https://github.com/CADDE-CENTRE linked to this publication.

Query 2.

And I do not see how the authors may improve this manuscript without adding some important data about morbidity among non-vaccinated and vaccinated persons.

Response 2.

Thank you for identifying this important limitation. We agree that as an ecological study design, individual level data about morbidity among vaccinated and non-vaccinated people could not be added. We have added this limitation to the discussion as follows

“Furthermore, vaccination status is not routinely recorded in aggregate case data, pre-cluding a more nuanced analysis including this variable.”

Query 3.

It would be useful to publish it because these data are from South American country but not very important.

Response 3.

Thank you - we agree that these data are relatively unique given their geographic extent and sample size.

Reviewer 2 Report

This is an excellent manuscript that will contribute significantly to our understanding of the seroepidemiology of COVID-19, particularly in the face of concurrent infection and vaccination.  This is an exceptionally well written manuscript and the presentation is absolutely clear.  The work presented is robust with regard to the methods and analyses used, large sample sizes, adjustments in datasets based on real world events, and description of study limitations.  This is one of those rare reviews where I do not have any substantive criticisms or suggestions to make.

Author Response

Response to reviewer 2

Query 1.

This is an excellent manuscript that will contribute significantly to our understanding of the seroepidemiology of COVID-19, particularly in the face of concurrent infection and vaccination.  This is an exceptionally well written manuscript and the presentation is absolutely clear.  The work presented is robust with regard to the methods and analyses used, large sample sizes, adjustments in datasets based on real world events, and description of study limitations.  This is one of those rare reviews where I do not have any substantive criticisms or suggestions to make.

Response 1.

We thank the reviewer for this encouraging (and indeed rare) review.

Reviewer 3 Report

The article "Predicting SARS-CoV-2 variant spread in a completely seropositive population using semi-quantitative antibody measurements in blood donors" analyses SARS-CoV-2 serology assay data on the brazilian population. The focus is not on the serology assay itself, but the data analysis, which is quite extensive and well made. The result are in line and confirming previous data and hypotheses. The study has several limitations, but the authors explain those in the discussion quite well. The conclusions feel a little exaggerated compared to the supporting data. But regardless, the article contains valuable insight and data about COVID.

Author Response

Response to reviewer 3.

Query 1.

The article "Predicting SARS-CoV-2 variant spread in a completely seropositive population using semi-quantitative antibody measurements in blood donors" analyses SARS-CoV-2 serology assay data on the brazilian population. The focus is not on the serology assay itself, but the data analysis, which is quite extensive and well made. The result are in line and confirming previous data and hypotheses. The study has several limitations, but the authors explain those in the discussion quite well. The conclusions feel a little exaggerated compared to the supporting data. But regardless, the article contains valuable insight and data about COVID.

Response 1.

Thank you for these helpful comments. We added some qualifying words to reflect the reviewer’s observation that the conclusions feel a little exaggerated (see abstract and discussion of re-submission).